# *MVsim* is a toolset for quantifying and designing multivalent interactions

Bence Bruncsics[1,3], Wesley J. Errington [2,3] & Casim A. Sarkar [2] ✉

Arising through multiple binding elements, multivalency can specify the avidity, duration, cooperativity, and selectivity of biomolecular interactions, but quantitative prediction and design of these properties has remained challenging. Here we present *MVsim*, an application suite built around a configurational network model of multivalency to facilitate the quantification, design, and mechanistic evaluation of multivalent binding phenomena through a simple graphical user interface. To demonstrate the utility and versatility of *MVsim*, we first show that both monospecific and multispecific multivalent ligand-receptor interactions, with their noncanonical binding kinetics, can be accurately simulated. Further, to illustrate the conceptual insights into multivalent systems that *MVsim* can provide, we apply it to quantitatively predict the ultrasensitivity and performance of multivalent-encoded protein logic gates, evaluate the inherent programmability of multispecificity for selective receptor targeting, and extract rate constants of conformational switching for the SARS-CoV-2 spike protein and model its binding to ACE2 as well as multivalent inhibitors of this interaction. *MVsim* and instructional tutorials are freely available at https://sarkarlab.github.io/MVsim/.

Multivalent interactions are fundamental building blocks of supramolecular systems. Deriving from multiple binding elements within sets of interacting molecules, multivalency is used to regulate intracellular compartmentalization[1–5], high-avidity interactions[6–9], ultrasensitivity[10], and dynamics and selectivity of molecular recognition[11–13]. The expansive utility of multivalency has driven multiple computational approaches to describe aspects of multivalent interactions[14–22]. These models range from fundamental treatments of linker-driven bivalent interactions[14,16], to system-specific descriptions of complex ligand recognition[17–19], to coarse-grained approaches that model multisite engagement at surfaces[20–22]. Given that multivalency is a powerful design element of natural and synthetic systems that is easy to implement yet difficult to predict, there is a need for holistic, quantitative molecular frameworks that can integrate existing approaches in the literature, that are extensible across the multivalency design landscape, that render molecular inputs and kinetic outputs in experimentally testable formats, and that can be presented as simple interactive tools for researchers performing model-guided experimentation. We previously developed a conceptualization of multivalency that described the noncanonical signatures of multivalent receptor–ligand interactions as the flux through an interconnected network of configurational microstates[23]. This approach provided highly resolved mechanistic insights into the dynamical events that underlie simple multivalent interactions and indicated a means with which to extend existing experimental techniques—such as surface plasmon resonance (SPR)—to macromolecular systems previously beyond the scope of quantitative analysis due to their complexity and heterogeneity[24,25]. However, this conceptual framework was limited to monospecific multivalent interactions between proteins of certain topologies and was also not practically implementable to quickly analyze and design a wide range of molecular systems[26–29].

[1]Department of Measurement and Information Systems, Budapest University of Technology and Economics, Budapest H-1111, Hungary. [2]Department of Biomedical Engineering, University of Minnesota, Minneapolis, MN 55455-0215, USA. [3]These authors contributed equally: Bence Bruncsics, Wesley J. Errington. ✉e-mail: csarkar@umn.edu

Here, we have developed an expanded computational method based on the original conceptual framework, which we present as *MVsim*, an interactive toolset with a simple graphical user interface (GUI) for the design, prediction, multidimensional parameter exploration, and quantification of multivalent binding phenomena. *MVsim* enables users to simulate multivalent binding through an expansive implementation of configurational multivalent networks within the MATLAB software environment[30]. With user-specified kinetic, topological, and structural parameters, *MVsim* simulates the conformational dynamics and binding responses for multivalent interactions that can be composed of multimeric, multidomain, and/or multi-ligand systems of interacting biomolecules. Further, *MVsim* synthesizes its outputs as sets of interactive kinetic traces to facilitate visualization, inspection, quantification, curve fitting, and experimental implementation of the structure–activity relationships and the information-coding intrinsic to multivalency.

We first validate the ability of *MVsim* to accurately simulate both monospecific multivalent interactions (i.e., a single repeated ligand domain on one binding partner and a single repeated target domain on the other) and multispecific multivalent interactions (i.e., more than one ligand domain and/or target domain in the interacting protein pair), the latter of which was not possible in our earlier model of multivalency[23]. To then demonstrate the application of *MVsim*, we use experiment-guided modeling to quantify switch-like signaling of synthetically designed systems[31], uncover design rules and predict the response dynamics of multivalent logic gates[31], and leverage multispecific ligand architectures to enable selective receptor targeting for therapeutic development[10]. Further, we apply *MVsim* toward the inspection and quantification of viral spike protein dynamics. At present, the importance of multivalency is assertively highlighted in the mechanics of infection and therapeutic targeting of the configurationally dynamic, trimeric SARS-CoV-2 S protein[32,33], its dimeric ACE2 receptor[34,35], and a growing library of designed monospecific and multispecific multivalent neutralizing inhibitors[36–40]. Here, we use *MVsim* to derive an effective concentration for the ACE2 interaction, quantify intramolecular rate constants of SARS-CoV-2 S protein receptor binding domain (RBD) conformational switching that enable host-cell engagement, and probe the consequences of variants with altered conformational stability of the S protein. This series of multivalent and multi-ligand simulations served as a means to quantify the relationships between macromolecular topology and SARS-CoV-2 S protein response dynamics, infectivity, and therapeutic targeting.

In sum, *MVsim* offers an intuitive and facile molecular design toolset, bringing enhanced quantification and predictive design of multicomponent and multivalent systems to protein engineering, molecular and cellular analysis, and therapeutic design.

## Results

### Development of *MVsim*

*MVsim* is a multivalent interaction toolset built upon our configurational microstate network model[23], which expanded upon prior modeling efforts in the literature by explicitly treating multivalency as a dynamic ensemble of binding configurations driven through local, topology-derived effective concentrations. *MVsim* represents a reconceptualization and application of the initial, more limited network model to now provide mechanistic descriptions of an array of biologically and therapeutically relevant multivalent systems and to quantitatively predict binding responses and conformational dynamics across a breadth of parameter space[31,37,40].

The creation of the *MVsim* toolset translated the fundamental concepts of the network model into the MATLAB coding environment through a series of implementations that represent significant advances in its ability to easily simulate a broad range of multivalent interactions. First, to describe a user-specified multi-ligand, multivalent interaction system, we developed a rule-based modeling routine to automatically enumerate all possible binding microstates and configuration transitions between them in order to generate a descriptive kinetic model (Extended Methods, Supplementary Information). For example, extended to its furthest, *MVsim* simulates competitive interactions among three topologically-varied trivalent ligands for a trivalent receptor, described with a system of 1538 differential rate equations.

Second, *MVsim* effectively and rapidly parameterizes the system of rate equations with computed topology-derived first-order rate constants of association. Here, *MVsim* uses dimensionally-reduced polar coordinate integrations of the molecular interaction volumes. With this approach, the frequency of all pairwise combinations of multivalent interaction between a ligand and receptor binding domain are calculated with joint probability density functions to yield a set of effective concentrations. This routine enables efficient calculations to be performed with high spatial resolution for nanoscale and mesoscale multivalent species with domain diameters, linkages, and persistence lengths exceeding 1000 Å.

Third, *MVsim* has an extensive multiparameter description of the molecular multivalent landscape that allows for zero-fit prediction of the response dynamics of fully parameterized systems where experimental multivalent data are absent. Conversely, *MVsim* enables parameter estimation for topologically under-characterized systems where multivalent binding kinetics have been measured. *MVsim* facilitates quantification of multivalent binding responses in terms of effective rate constants of association ($k_{on}^{eff}$) and dissociation ($k_{off}^{eff}$), equilibrium dissociation constants ($K_D^{eff}$), competitive inhibitor potency ($IC_{50}$), and Hill coefficients ($n_H$) describing ultrasensitive switch-like behavior.

### Parameter inputs

Interfacing *MVsim* with MATLAB's app design environment enabled the creation of a tabbed GUI to guide the specification of biologically important manifestations of multivalency via multiparameter inputs. The *MVsim* GUI enables full input parameterization of the domain and linkage topologies of the ligand(s) (Fig. 1a) and receptor (Fig. 1c), the monovalent kinetics between each pairwise combination of ligand-receptor binding domains (Fig. 1a), the temporal ligand concentration dynamics (Fig. 1b), and the set of parameters that govern SPR and related kinetic studies, including the association and dissociation times, flow rate, and level of the immobilized receptor. In addition, for instances where detailed topological information is known, *MVsim* allows users to directly input effective ligand concentrations and end-to-end probability density functions for the multivalent system of interest (Supplementary Fig. 1). Finally, once a multivalent design has been inputted, the user can initiate *MVsim* and subsequently survey a range of non-topological parameter variants in quick succession (Fig. 1d). Tutorials for navigating the GUI and inputting parameters are available in the Supplementary Information and on the *MVsim* GitHub page (https://sarkarlab.github.io/MVsim/). Supporting figures that detail the parameter inputs used for each simulation in this study are also provided in the Supplementary Information.

### Simulation outputs

Following the initiation of a simulation, *MVsim* provides users a variety of means to visualize, interact with, and export the simulated response kinetics. Most simply, the simulation results are displayed within the output field as an interactive plot of the binding response signal as a function of the specified association and dissociation time (Fig. 2). Here, users can choose between two graphical outputs of the response kinetics. First, as is typical of experimental kinetic binding data, a plot of a user-specified ligand concentration is displayed (Fig. 2a). Alternatively, users can select a plot of all composite microstate configurations underlying the response signal (Fig. 2b), binned according to valency class (Fig. 2c) or by ligand class to observe the competitive

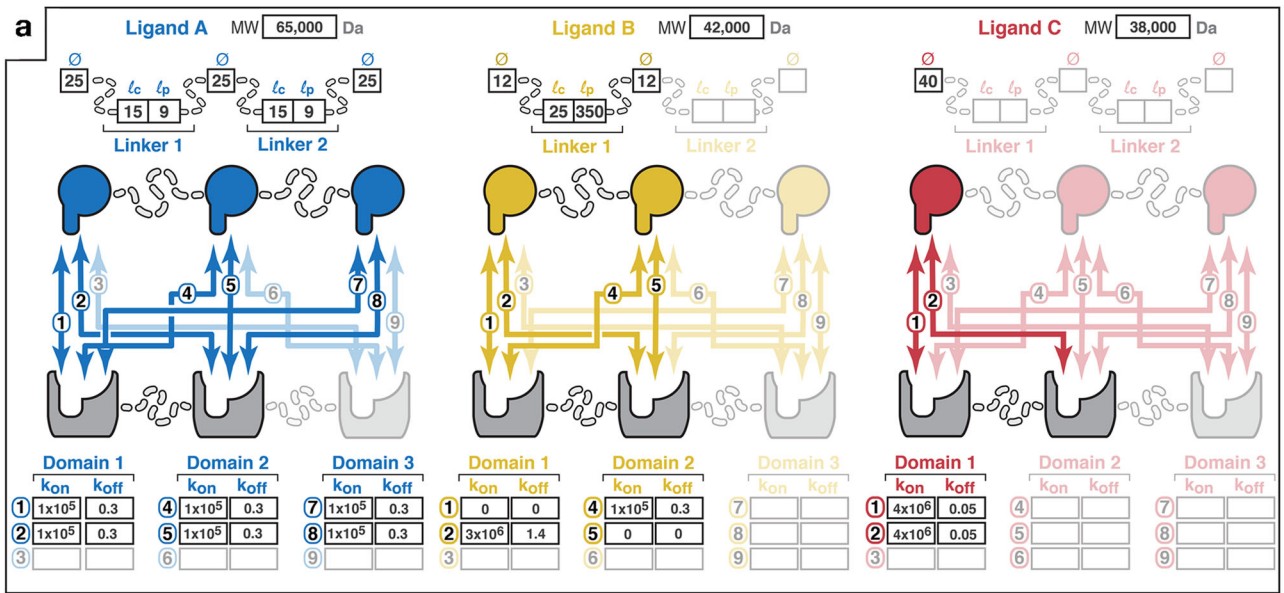

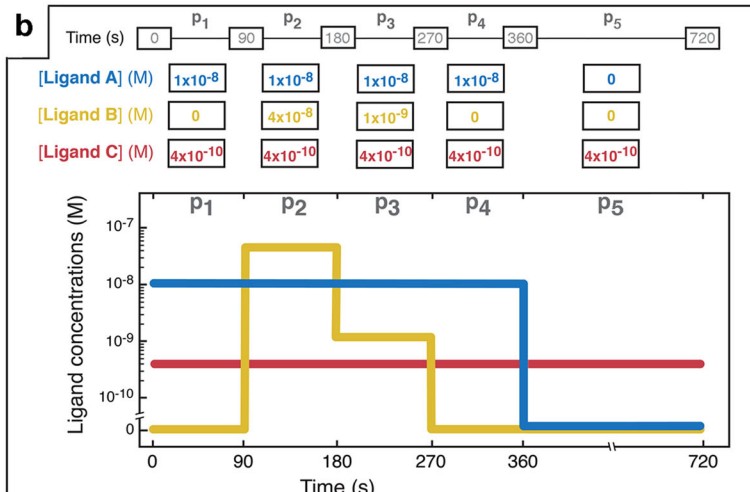

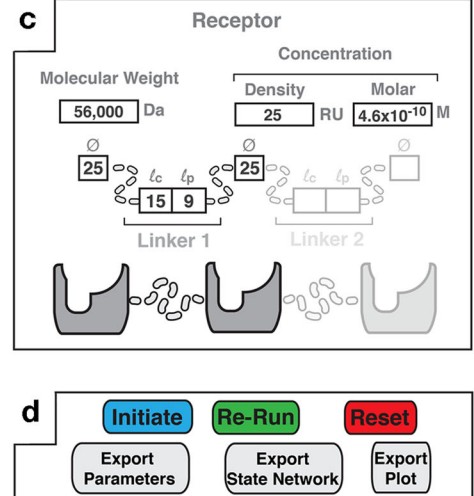

**Fig. 1 | The *MVsim* input design interface provides interactive parameter specification for systems of multivalent, multi-molecular interaction. a** A point-and-click interface enables the user to select the number of ligands (up to three) and valencies of the ligand(s) and receptor (up to trivalent) that compose the multivalent system. Based upon the chosen design, the user specifies the structure of each of the ligands by entering the applicable molecular weight (MW); the binding domain diameters (Ø); the contour lengths ($l_c$) of the linkers (i.e., the maximum end-to-end distance; e.g., 3.5 Å and 1.5 Å per amino acid for a random coil and alpha helix, respectively); and the persistence lengths ($l_p$) of the linkers. Further, the applicable combinatorial interactions (numbered 1 to 9) unique to each receptor–ligand pairing are highlighted. Parameter fields allow the input of monovalent rate constants for each pairwise interaction. Non-binding interactions can be indicated with $k_{on}$ and $k_{off}$ values of zero (e.g., as illustrated with Ligand B in yellow for interactions "1" and "5"). **b** An input field allows the user to specify patterns of the total, bulk ligand concentrations. An association phase occurs during periods of non-zero bulk ligand concentration (e.g., 90–270 s for Ligand B). Dissociation phases occur when the ligand is removed from the bulk solution (e.g., 360–720 s for Ligand A). Here, Ligand C is specified as continuously present in solution during the 720 s of the interaction timecourse. The graphical display allows visualization of the specified bulk concentration pulse pattern. **c** User input parameters for the receptor. Receptor concentration can be specified as either an SPR-mimicking surface density (measured in RU; where 1 RU equals -1 pg/mm²) or a molar concentration. Receptor topology is specified in the same form as described above for the ligands. **d** The *MVsim* controller tab enables initiation, iteration, and export of binding simulations. "Initiate" executes a simulation. "Re-run" executes an abbreviated simulation used when no changes were made to the valency or topology of the system. "Reset" relaunches the app and clears user input parameters from all fields. Tutorials illustrating the input interfaces are available in the Supplementary Information and at https://sarkarlab.github.io/MVsim.

binding dynamics among multiple ligands (Fig. 2d). *MVsim* additionally enables users to visualize the dynamic evolution of the microstate network via an interactive map (Fig. 2e) and to export the response kinetics as a set of tab-delineated text files to facilitate deeper analysis through offline plotting and curve fitting, and through microstate network analysis within the Cytoscape software environment[41]. *MVsim* also allows users to directly inspect both the computed probability density functions and effective concentrations (Supplementary Fig. 2).

**Assessing performance against experimental model systems**

To evaluate the accuracy of simulating complex topologies, *MVsim* was used to predict the binding response dynamics for three multivalent systems (Fig. 3). First, we evaluated our previously constructed monospecific multivalent interaction (i.e., composed of a single pair of protein interaction domains; Fig. 3b). Then, we evaluated two additional experimental systems: a multispecific multivalent interaction (i.e., composed of two sets of interaction domains;

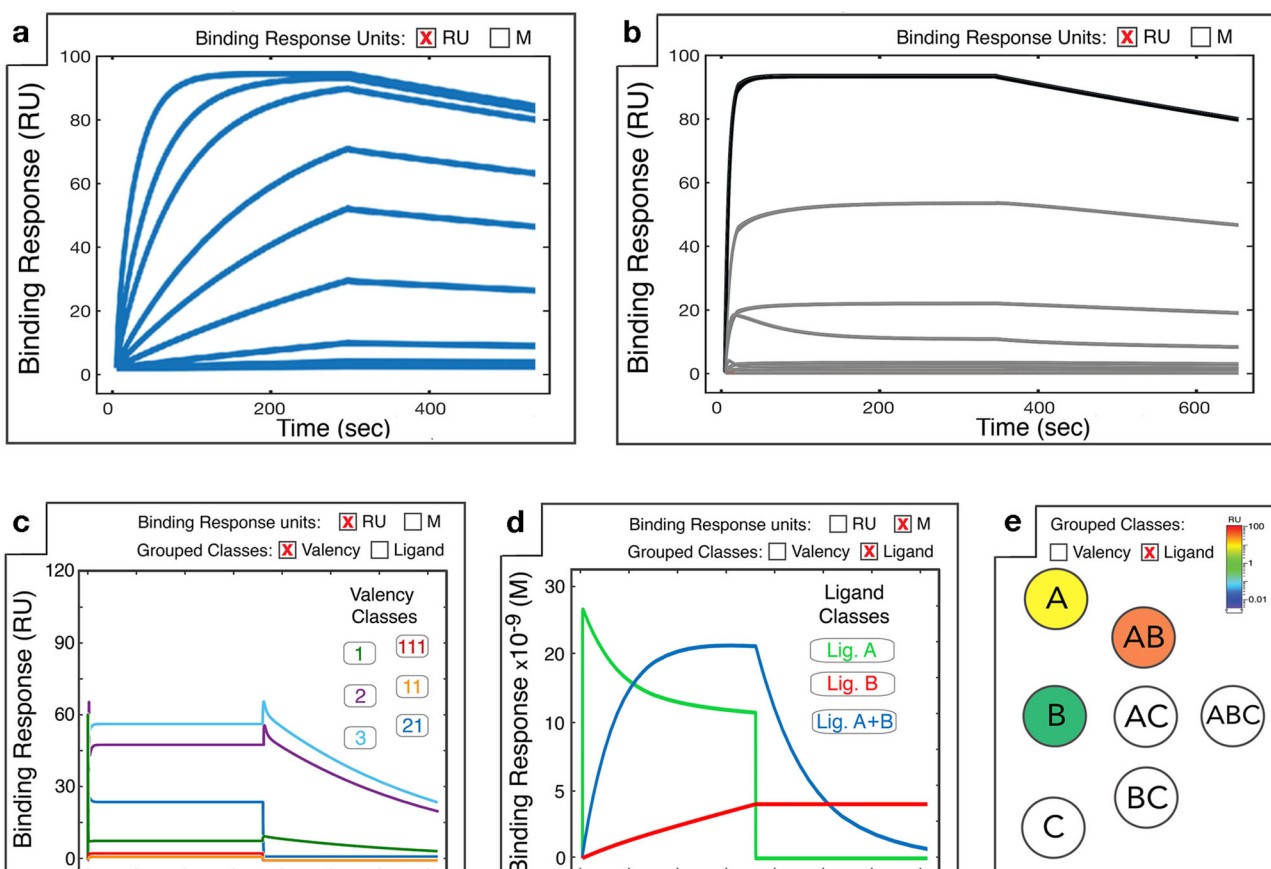

**Fig. 2 | *MVsim* generates a series of outputs that enable interactive visualization of binding responses, configurational microstates, and multi-ligand dynamics.**
**a** A simulated SPR sensorgram displays the overall response dynamics (i.e., summation of all ligands and binding microstates) for specified ligand concentration(s). Indicated here are the binding responses for a serial dilution of a single ligand binding to a receptor with association (0–300 s) and dissociation (300–600 s) phases. For a simple quantitative comparison between simulations, an overall effective $K_D$ can be calculated by the equilibrium method.
**b** For a specified ligand concentration, all composite microstates are displayed. **c**, **d** To facilitate analyses of the binding responses, the microstates can be binned according to either **c** valency or **d** ligand class. **e** For visual analysis of the evolution of a network of microstates in **b**, an interactive graph shows population changes in microstate classes over a timecourse of association and dissociation. The simulations in these figure panels are not related to one another; they are for illustrative purposes to highlight the distinctive features in each view. Tutorials illustrating the output interfaces and features are available in the Supplementary Information and at https://sarkarlab.github.io/MVsim.

Fig. 3d) and a combined multispecific/multi-ligand interaction (Fig. 3g). To directly compare *MVsim* to our previous studies, we evaluated a monospecific interaction composed of a kinetically and structurally parameterized pair of protein–protein interaction domains (SH3 and SH3-binding peptide (SBP); our experimental parameterization by SPR is shown in Fig. 3a)[42], rendered multivalent with polypeptide linkages parameterized using literature-derived topological values (Fig. 3b)[43–46]. Here, *MVsim* predicts monospecific multivalent binding (Fig. 3c) but now also shows improved sensitivity to the topological constraints that can impede certain configurations, such as those that requiring contorted twisting of interdomain linkages, resulting in model-experiment agreement with a root-mean-square error (RMSE) of 2.1 RU (7% of the average signal; Fig. 3c). Moreover, by extending these simulations through systematic parameter variation, *MVsim* identified ligand concentration as the parameter with the most sensitive effect on the simulated binding response of this system. By increasing the simulated ligand concentration just twofold, an improved overall model-experiment agreement was observed in both the low concentration ligand conditions and in the magnitude of the biphasic association. This improvement in fit, however, is at the expense of capturing the modest kinetic burst at early times (Supplementary Fig. 3a),

underscoring the complex, interconnected relationships between a single parameter value and multiple descriptive noncanonical features of multivalent binding responses.

We further used *MVsim* to make predictions of multispecific and multi-ligand interaction systems. In the first validation, a multispecific receptor–ligand architecture was designed through incorporation of a second set of protein–protein interaction domains (Prb and Pdar; Fig. 3a). Again, in addition to the experimentally determined monovalent kinetic rate constants (Fig. 3a), literature-derived properties of the molecular structures and topologies were used to parameterize the model (Fig. 3d)[43–46] and to generate a simulated dataset (Fig. 3e). Comparing simulation with the corresponding SPR dataset (Fig. 3f) demonstrates good agreement with regard to the ability of *MVsim* to predict a priori the magnitude and multiphasic character of the experimental binding responses. Further, *MVsim* provides mechanistic explanations for these binding responses, showing, for example, the contribution of high-stoichiometric configurations to the microstate ensemble driven by the use of rigid, α-helical linkers (Supplementary Fig. 4a–c). The quantitative RMSE between model and experiment of 1.6 RU (6% of the average signal) indicates that the simulations effectively captured the experimental binding responses to a significant degree. Moving beyond zero-fit predictions, *MVsim* identifies

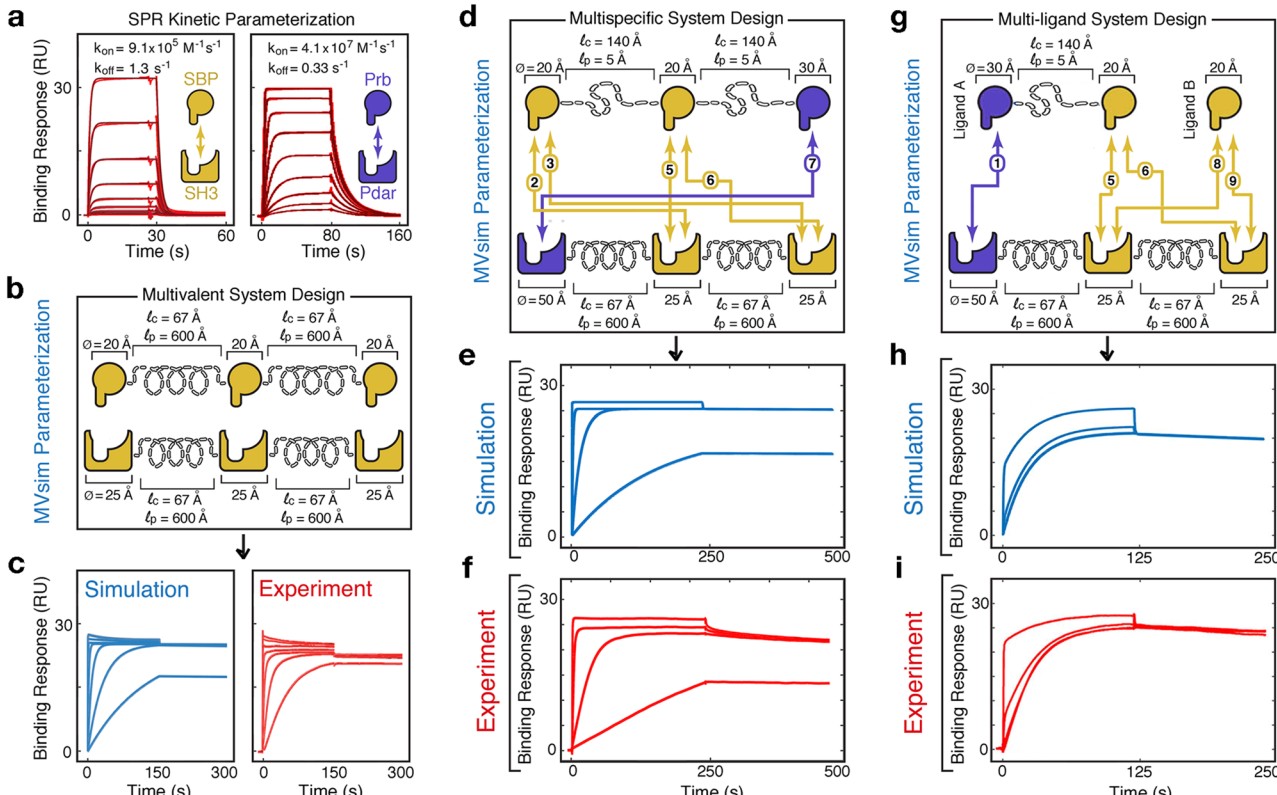

**Fig. 3 | *MVsim* accurately simulates beads-on-a-string multivalent, multi-specific, and multi-ligand interactions. a** Monovalent SPR kinetic rate constants were experimentally determined for the SH3-binding peptide (SBP)-SH3 (**a**, left panel) and Prb-Pdar (**a**, right panel) interactions that were used to build the multivalent systems. Kinetic fits with a "rapid mixing" 1:1 Langmuir model showed good agreement since the experimental conditions were not significantly mass-transfer limited (see Supplementary Experimental Methods). **b** A trivalent, monospecific receptor–ligand interaction was engineered and parameterized within *MVsim* using values for the kinetic rate constants of association ($k_{on}$) and dissociation ($k_{off}$), domain diameters (Ø), and contour ($l_c$) and persistence ($l_p$) lengths for the linkers. **c** Simulated (**c**, left panel) and experimental (**c**, right panel) binding response dynamics for the trivalent, monospecific interaction at seven ligand concentrations (0.98, 3.9, 15.6, 62.5, 250, 1000, and 2000 nM). Model-experiment RMSE is 2.1 RU. **d** A trivalent, bispecific receptor–ligand interaction was engineered and parameterized using values for the kinetic rate constants of association ($k_{on}$) and

dissociation ($k_{off}$), diameters (Ø) for the protein–protein binding domains, and contour ($l_c$) and persistence ($l_p$) lengths for the alpha-helical linkers. Arrows indicate compatible interactions between receptor and ligand binding domains. **e** Simulated binding responses for the parameterized trivalent, bispecific interaction at four simulated ligand concentrations (0.1, 1, 10, and 1000 nM). **f** Experimental SPR binding response dynamics for the trivalent, bispecific interaction at the same four ligand concentrations as in **e**. Model-experiment RMSE is 1.6 RU. **g** The Pdar-Prb and SBP-SH3 protein–protein binding domains were used to create a multi-ligand system. **h** Simulated binding response dynamics modeled by *MVsim* for the parameterized dual ligand system. An overlay is shown of binding responses for three simulated mixtures of ligands A and B (1 nM A + 2.5 nM B; 1 nM A + 50 nM B; and 1 nM A + 250 nM B). **i** Experimental SPR binding response dynamics for the same three dual ligand mixtures as in **h**. Model-experiment RMSE is 2.0 RU. The input parameters for these simulations are provided in Supplementary Fig. 5. Source data are provided as a Source Data file.

parameters that most sensitively affect the multivalent binding response. Here, for example, the persistence lengths of the ligand linkages present as the most sensitive system parameters. Increases to the rigidity of the ligand linkers in this system served to place the ligand-receptor binding domains slightly out of optimal register, modestly enhancing the presence of high-stoichiometric binding states in the association phase and increasing the rate of dissociation (Supplementary Fig. 3b).

As a second model-experiment validation, the monospecific and multispecific designs (Fig. 3a, b, d) were combined to create a kinetically and topologically parameterized dual ligand interaction system (Fig. 3g). The simulated binding responses (Fig. 3h) succeed in capturing the multiphasic association and dissociation dynamics present in the experimental SPR data (Fig. 3i). Moreover, beyond simply predicting the overall kinetics of the system, *MVsim* provides insights into the mechanics of multiple multivalent and multispecific ligands competing for a receptor, and attributes these molecular properties back to the macroscopically observable features of the multiphasic binding responses. Here, for example, *MVsim* captures how effective rate constants of dissociation can be dictated by valency and can be used to effect the temporal ordering of

interactions between a rapidly, but more transiently, binding monovalent ligand and a slower, but more avid, multivalent ligand (Supplementary Fig. 4d–f). Further, as in the multispecific validation, parameter variation can again be used to assess model-experiment agreement. Here, for example, our zero-fit simulation of this multivalent, bispecific, and multi-ligand interaction system agreed with the experiment with an RMSE of 2.0 RU (9% of the average binding signal). While the zero-fit simulation again captured the biphasic features of the association and dissociation phases of the binding response, the relative proportions of the fast and slow phases were less well predicted. Through parameter variation, *MVsim* identified $k_{on}$ for Ligand B as a parameter that sensitively affects these multiphasic binding responses (Supplementary Fig. 3c). Here, for example, a twofold increase in the Ligand B $k_{on}$ in the model yielded better quantitative agreement to the fastest phase of the experimental association curve at the highest ligand concentration (Supplementary Fig. 3c; 0–2 s of the association phase) and the slowest phase of the experimental dissociation curve (Supplementary Fig. 3c; 125–250 s). However, these two improvements in model-experiment agreement came at the expense of over-representing the amount of Ligand B that remains bound to the receptor at equilibrium

(Supplementary Fig. 3c; -125 s). Again, this analysis of parameter sensitivity highlights the complex relationship between a single parameter value and the descriptive features of the binding response, here specifically in regard to the competitive dynamics between the ligands for the receptor.

## Applications to multivalent system design and quantification

*MVsim* was established to both guide the design and implementation of multivalent properties and to facilitate parameter estimation for existing and incompletely characterized natural and synthetic multivalent systems. Here, the model's lack of reliance upon fitted parameters enables *MVsim* to better describe the additive, competitive, and cooperative relationships implicit among kinetic, topological, and valency parameters and to apply these to the quantification of multivalent properties, such as effective concentration, avidity, and binding selectivity. To evaluate the performance of *MVsim* as a molecular design and quantification tool, we assessed its ability to design and predict binding response dynamics in four different instances and applications of multivalency.

## *MVsim* predicts ultrasensitive behavior in engineered protein switches and logic gates

The effective concentration that drives multivalent binding gives these systems the inherent ability to produce nonlinear input/output response dynamics. It has been previously demonstrated, for example, that ultrasensitive toggling can be driven through the introduction of monovalent counterparts into a multivalent system[31]. Dueber et al. showed that cooperative competitive dissociation of multivalent protein–protein complexes effects switch-like transitions that can be leveraged to control the fractional saturation of receptor–ligand interactions and enzymatic activity. Here, we apply *MVsim* to study the activation dynamics of engineered bivalent and trivalent protein switches and identify critical parameters for optimal system performance. *MVsim* quantitatively predicts the relationship between the valency of the system and the magnitude of its cooperative transition to an active state (Fig. 4a). The functional range of multivalent switches can be extended through the incorporation of multispecific interactions. This design approach enables the creation of AND logic gates in which a switch response is elicited only by a programmed combination of molecular inputs. Even though catalytic activity (and not SPR) was used to measure output in this experimental system, *MVsim* could still qualitatively capture the three-input gating function for the available experimental data (Fig. 4b). In addition, our simulations suggested likely spurious two-input activation of the system (Supplementary Fig. 6a, b; bars v–vii), as a consequence of the nontrivial activation that is observed computationally by single inputs (Supplementary Fig. 6b; red bars ii–iii) and that is even more prominent in the experiments, due to significant basal activation and single-input PDZ activation (Supplementary Fig. 6b; gray bars i and iv, respectively). The simulations failed to predict these latter two instances, which were the mildest activators (i.e., no input or the lowest-affinity input), suggesting that the sterically constrained experimental system produces basal activation beyond the idealized topological treatment in the simulations. Finally, through parameter exploration, *MVsim* guided the identification of key steric constraints within the system that could contribute to the observed basal and single-input activations and indicated an optimized molecular design (Supplementary Fig. 6c) that could minimize these impediments within the auto-inhibited conformation with extended flexible linkages and equalized the interdomain binding kinetics to improve the dose-responsiveness of the agonists. The result was a more tightly controlled simulated system that better maintains an inhibited state in the presence of zero-, one-, and two-input conditions (Supplementary Fig. 6c; bars i, ii–iv, and v–vii, respectively).

## *MVsim* informs the use of multispecificity for molecular recognition and therapeutic targeting

Multispecificity is a potent molecular design element that is widely used in drug discovery and cell engineering. By leveraging two or more distinct binding epitopes, multispecific interactions are employed to engineer highly avid and selective molecular recognition for use in such applications as bispecific therapeutic antibodies[10,27] and chimeric antigen receptor T cells[47]. Multisite recognition additionally enables higher-order information processing, allowing these multispecific systems to generate differential outputs to varying combinations of inputs[11,12]. Because the network model of multivalency computes multivalent binding as the cooperative sum of its composite interactions, *MVsim* is well-suited to the study of such multipartite interactions.

For example, multispecific interactions can be designed to maximally exploit any degree of variation in the type and number of surface receptors and antigens within a population for the purposes of selective targeting[10]. In this regard, we directed *MVsim* to address a design question: given a population of three distinct types of antigenic cell surfaces (Fig. 4c), what are the optimal ligand designs that can singly, doubly, and triply interrogate the population? *MVsim* demonstrates that the composition of the target receptor serves as a generally useful guide for ligand design, as seen, for example, in the relative selectivites of mono-, bi-, and trispecific Ligands 1, 3, and 7, respectively, for Receptor 3 (Fig. 4c). Moreover, selective recognition can be further tuned using designed linkages that leverage the spatial proximity between receptor target surfaces; Ligand 2b (rigid linkage) has greater selectivity than Ligand 2a (flexible linkage) for Receptor 2 (Fig. 4c).

The information-coding capacity of multivalent interactions can also effect the temporal ordering of ligand binding to a single receptor target when multiple multivalent ligands are introduced simultaneously (Supplementary Fig. 7a–c), a phenomenon that is not possible in a comparable monovalent system (Supplementary Fig. 7d–f). Exploration of the simulated parameter space in this system revealed a multivalent design leveraging kinetics, avidity, and stoichiometry that enables serial phases of dominant ligand engagement by exploiting the cumulative effects of concurrent binding afforded by multispecificity, the cooperative, competitive binding of multi-ligand dynamics, and the generation of effective dissociation rate constants via multivalency (Supplementary Fig. 7c).

## *MVsim* models the multivalency and avidity of SARS-CoV-2 S protein interactions

At present, one of the most prominent and consequential displays of multivalent binding involves the surface spike (S protein) of SARS-CoV-2. The S protein is a sophisticated, conformationally activated molecular system that mediates selective recognition of target cells and generates the driving force needed to overcome the energy barrier of membrane fusion, thus enabling viral entry into the host[32,33]. The multimeric and multivalent configuration of the S protein is central to these functions[48]. Trimeric assembly serves to stabilize the S protein against erroneous fusogenic conformational changes, establish allosteric control, and potentially present multiple receptor binding domains (RBDs) that bind multivalently with a host-cell surface populated with dimeric ACE2 receptor proteins[48]. In response to these natural displays of multivalency, this same principle has been mobilized in therapeutic designs intended to neutralize, inhibit, or otherwise uncouple the structure–activity relationship of the S protein[35–40].

Despite the significantly more complex multivalent architecture of the S protein compared with our previously described applications, *MVsim* can be effectively parameterized to model and quantify critical structural properties of the S protein-ACE2 interaction (Supplementary Fig. 8a, b). For example, it remains an open question the extent to which the trimeric S protein can multivalently engage a bivalent ACE2 receptor. This is of considerable importance for our understanding of

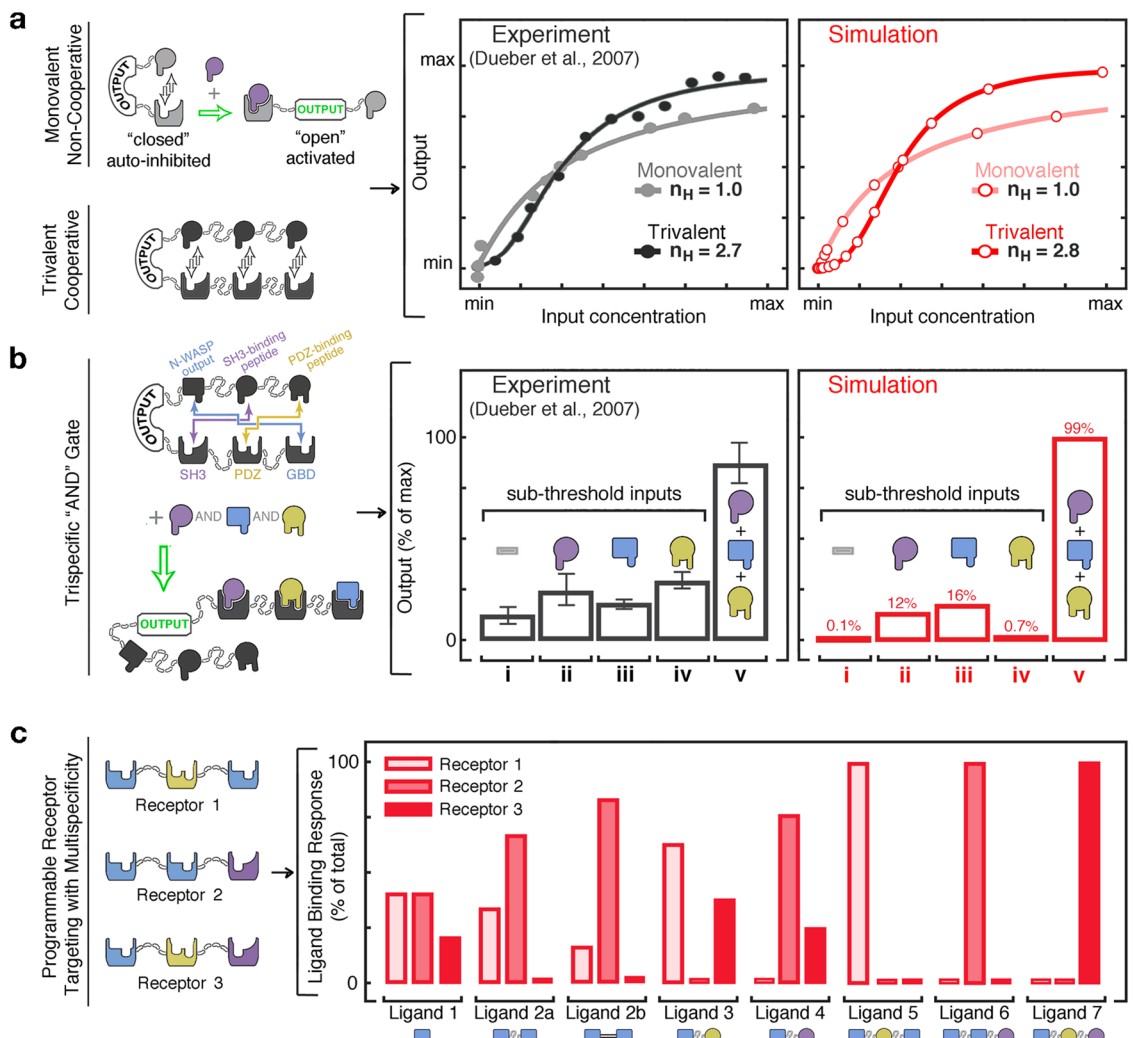

**Fig. 4 | *MVsim* predicts and informs the design of switch-like dynamics, logic operations, and target-receptor selectivity implicit to multivalent systems.** **a** Experimental response dynamics of synthetic monovalent and trivalent switches from ref. 31 were used to benchmark the predictive performance of *MVsim* simulations described by the reported structural, topological, and kinetic parameters. Ultrasensitivity of each simulated response is reported with a calculated Hill coefficient ($n_H$) for direct comparison with the reported literature values. **b** Experimental output responses for a trispecific AND logic gate, also from ref. 31, benchmarked against an identically parameterized system in *MVsim*. For clarity, the AND gate is depicted (diagram, top left) in its originally designed twisted configuration (detailed in Supplementary Fig. 6). Simultaneous addition

of the three inputs (SH3, PDZ, and GBD-binding peptides; colored purple, yellow, and blue, respectively) flips the AND gate into an active conformation. To ensure optimal system performance, it is desirable to prevent activation of the AND gate with subthreshold inputs, though this is difficult to achieve through ad hoc experimentation. Here, nontrivial subthreshold activation is indeed observed (bars i–iv). **c** *MVsim* specifies optimal design of multivalent and multispecific ligands to yield desired patterns of selective interactions within a pool of three receptors with common binding domains. The affinities for the receptor–ligand binding domains (colored blue, yellow, and purple) are as described in **b**. The *MVsim* input parameters for the simulations in **a**–**c** are further detailed in Supplementary Fig. 6.

how the affinity and avidity of S protein binding relate to infectivity, and what consequences this poses for therapeutic inhibition[40,49]. In synthetically engineered multivalent instances of the RBD-ACE2 interaction, *MVsim* quantitatively predicts the relative lack of steric hindrance that affords the ultra-high-avidity binding observed in a study by Chan et al.[40] (Fig. 5a, b). In contrast, experiments performed on more biologically mimetic S protein-ACE2 interactions indicate a significant impediment toward high-avidity binding[40]. Using *MVsim* to fit therapeutic neutralization datasets reveals an effective ligand concentration, [$L_{eff}$], for the second engagement event between S protein and ACE2 that is 2000-fold less potent than that observed in the sterically unimpeded system (Fig. 5c, d)[40]. This inability to achieve high-avidity binding (e.g., a network in which >95% of the populated microstates are bound with maximal valency, as is the case for the "High" simulation in Fig. 5d) can be explained by the combination of the rigidity of the ACE2 dimer and the apparently constrained,

directional motion of the linkage tethering the RBD. Quantitative modeling approaches such as these indicate a significant potential for therapeutic designs that can potently outcompete the RBD-ACE2 interaction by leveraging multivalent binding in ways inaccessible to the S protein (Fig. 5e, f). Specifically, *MVsim* predicts that up to 1000-fold enhancements in $IC_{50}$ values can be achieved through the use of topologically precise and constrained linkages within a designed, trivalent multispecific neutralizing therapeutic (Fig. 5e, f). Reciprocally, *MVsim* further demonstrates how bivalency can be effectively leveraged with appropriate linkages to avidly block the RBD binding surfaces of the ACE2 dimer (Supplementary Fig. 8c, d).

### *MVsim* quantifies the dynamics of SARS-CoV-2 S protein conformational switching

In addition to the sterically impeding immobility of the S protein-ACE2 interaction (Fig. 5c, d), multivalent engagement is limited by the

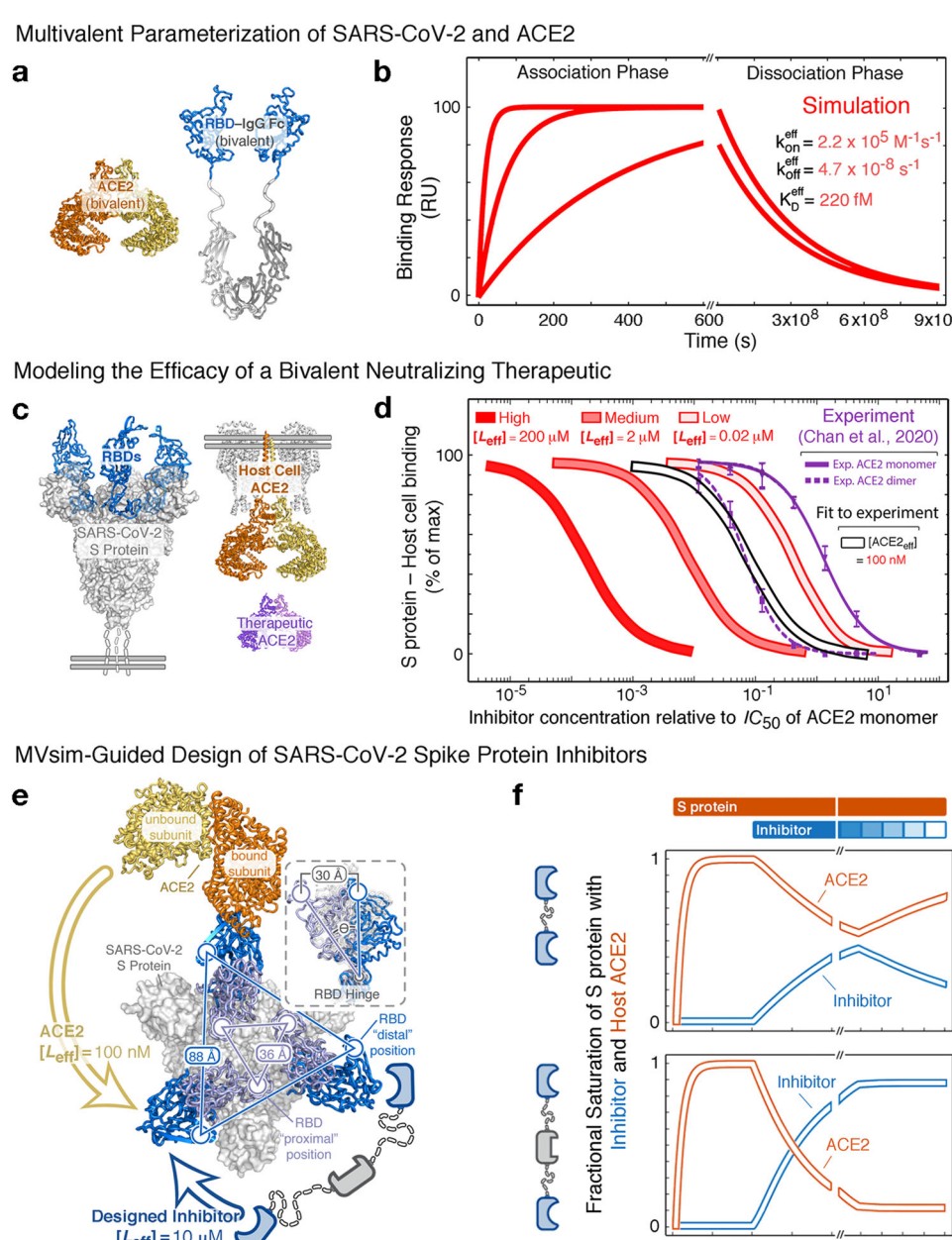

**Fig. 5 | *MVsim* models the multivalency of the SARS-CoV-2 S protein RBD and ACE2 interaction and suggests therapeutic design strategy. a** An idealized, flexible synthetic design of an ACE2–RBD bivalent architecture. Here, the synthetic design removes the RBD from the biologically relevant and constrained context of the rest of the S protein. **b** The flexible RBD linkers afford a high-avidity bivalent interaction with the dimeric ACE2 that was beyond the quantification limits of the experimental SPR. Here, *MVsim* was parameterized with the features of the experimental systems and offers prediction and quantification of the ultra-high-avidity interaction. **c** Application of *MVsim* to a biologically relevant instance of the SARS-CoV-2 S protein RBD and ACE2 interaction. Here, the therapeutic neutralizing activity of soluble, dimeric ACE2 (purple) was quantified in a SARS-CoV-2 pseudo-virus-host-cell system[40]. **d** The resulting $IC_{50}$ datasets were applied to *MVsim* in order to fit for a more biologically relevant determination of the multivalent binding capacity of the S protein-ACE2 interaction. The experimental data (purple traces) are adapted from ref. 40 The best fit from *MVsim* gave an $[L_{eff}]$ of 100 nM (curve outlined in black), falling between the "Low" and "Medium" standard curves (shades of red), which represent no capacity and a modest steric capacity for bivalent binding, respectively. These simulations indicate that the RBDs in the full context of the S protein are significantly impeded for direct bivalent binding to ACE2. **e** This steric impediment can be exploited to maximize neutralizing potency by fully leveraging therapeutic multivalency. **f** *MVsim* can test the design of neutralizing inhibitors that maximally outcompete the ACE2 interaction. Designs leveraging monospecific bivalency (top panel) and trivalent bispecificity (bottom panel) are computationally modeled for their neutralizing strengths and off-rate dependent pharmacokinetic half-lives in the presence of constant S protein (orange bar above plot) and decaying concentration of therapeutic (blue bar). The input parameters for these simulations are given in the *MVsim* user tutorial in the Supplementary Information.

accessibility of the RBDs, as they dynamically sample configurations ranging from the occluded yet stabilized "RBD-down" conformation to the labile yet ACE2-binding competent "RBD-up" conformation[33,48]. The dynamics of this range of RBD motion are a significant target of selective pressure as the benefits of maximizing host-cell binding are countered by the need to stabilize the S protein against spontaneous fusogenic conformational change and immune surveillance of exposed critical surfaces[48]. To examine intramolecular conformational changes that yield multivalent binding, we applied *MVsim* to simulate a multicomponent experimental system consisting of a stabilized

trivalent S-protein, a set of first-order rate constants ($k_{up}$ and $k_{down}$) describing RBD conformational change, and a trivalent ligand specific for the RBD-down conformation (Fig. 6a). *MVsim*, constructed and parameterized in this way, succeeded not only in recapitulating the experimental multiphasic kinetic traces obtained in a study by Schoof et al.[37] (Fig. 6b, c), but also in relating these response dynamics to the rates of RBD conformational switching. *MVsim* was used to derive the best-fit parameter values that describe the conformational switching: $k_{up} = 0.017\,s^{-1}$ and $k_{down} = 0.008\,s^{-1}$ (Fig. 6d, e). These correspond to individual RBD half-lives of ~1.4 min in the RBD-up configuration and ~0.7 min in the RBD-down configuration for this stabilized, in vitro S protein system. The data-richness of multiphasic SPR sensorgrams is underscored by the fact that the S protein binding response dynamics are uniquely determined by a single set of kinetic constants (Fig. 6d, e). For example, variations in $k_{up}$ and $k_{down}$ that maintain the equilibrium constant (i.e., constant $k_{up}/k_{down}$ ratio) nonetheless result in diagnostically different response dynamics (Supplementary Fig. 9). To further assess the accuracy of the *MVsim*-derived values of $k_{up}$ and $k_{down}$, these rate constants were used to parameterize a simulated SPR experiment probing the lifetime of the stabilized RBD-all-up state. Here, good agreement was observed between model and experiment (Fig. 6f, g)[37].

Given the uniqueness of the fitted parameters in this system, *MVsim* should be similarly capable of uncovering the conformational dynamics of other S proteins. Notably, the half-lives for the synthetically stabilized S protein examined in this study are ~35-fold slower than those recently measured for the native S protein using FRET sensors[50]. In addition, S protein conformational dynamics are of particular importance for understanding the mechanisms through which emerging SARS-CoV-2 mutational variants of concern (VOCs) increase infectiousness[51]. Mechanistically, S protein VOCs can function to stealth this protein from host immune surveillance, enhance the binding kinetics/affinity of the RBD-ACE2 interaction, stabilize the RBD-up configuration to increase the avidity of virion-host-cell engagement, and/or augment conformational allostery that enables RBD binding to prime activation of membrane fusion[51]. To further apply *MVsim* to study S protein conformational dynamics, simulations were performed with the parameterized S protein RBD ensemble (Supplementary Fig. 10a) to probe the effects of $k_{up}$ and $k_{down}$ on the ensemble of RBD configurations (Supplementary Fig. 10b–d). Stabilization of the RBD-up state leads to commensurately stronger ACE2 receptor binding (Supplementary Fig. 10e–g).

## Discussion

*MVsim* is a toolset created for the design, prediction, quantification, and mechanistic analysis of multivalent molecular interactions. It empowers users to explore topologically complex multicomponent systems with an interactive GUI and to probe the relationships among configurational dynamics, cooperativity, effective concentration, and competitive binding that underlie the programmability of multivalent behavior. *MVsim* offers a considerable range of user inputs to parameterize the composition, kinetics, structure and topology, conformational flexibility, and component concentrations to simulate various disparate instances of multivalency in natural systems and synthetic designs.

Effectively simulating complex instances of multivalency has been hindered by the inherent combinatorial and spatial complexities that arise from binding domains sampling increasingly large, sterically constrained volumes to engage in a multitude of transient, pairwise interactions with unique energetic permissibilities[7,8,10]. *MVsim* addresses this challenge by combining rule-based modeling and multidimensional integrations to rapidly simulate system behavior by effectively tracking the evolution of hundreds of configurational binding state transitions throughout the lifetime of the molecular interaction. This generalized and extensible computational approach

provided a means to create a consolidated modeling routine that yielded accurate and meaningful predictions of systems including simplistic beads-on-a-string topologies, intramolecular switches, and conformationally-regulated multicomponent assemblies.

Here specifically, we demonstrate the ability of *MVsim* to capture the unique multiphasic kinetics characteristic of multi-ligand, multi-specific, and multivalent systems. We further show how *MVsim* can be used to predict and refine the design of systems that leverage multivalency to achieve nonlinear and ultrasensitive outputs, as well as the additional layering of multispecificity to create AND gate input/output operations. Further, we show use of *MVsim* in the advanced application of multispecificity toward the design of multivalent ligands capable of maximally distinguishing among pools of receptor targets. Finally, to demonstrate the features and multiparameter inputs of *MVsim* applied to their fullest extent, a variety of therapeutically relevant structural features were computed for the SARS-CoV-2 S protein. Notably, *MVsim* was used to extract the effective concentration for the ACE2 interaction, quantifying the sterically unfavorable interaction that had previously been inferred from structural modeling, and to devise therapeutic designs that more effectively leverage multivalent binding. In addition, *MVsim* was readily applied to bulk kinetic measurements to extract conformational rates of RBD switching, a biophysical property of the S protein and other viral spike proteins that are typically measured using single-molecule FRET[50,52]. In contrast to these sophisticated probe-based techniques, *MVsim* can more easily provide mechanistic insights into the relationship between S protein sequence and its structural dynamics. For example, with meaningful parameterization, *MVsim* could aid in mechanistically parsing the multiple virulence-enhancing features that comprise most VOCs and to quantify the consequences of a mutational profile that simultaneously alters the RBD ensemble, enhances the kinetics of ACE2 binding, and reduces the potency of a neutralizing therapeutic.

The modular construction of *MVsim* also enables its straightforward extension to additional instances of multivalency. For example, additional configurational network tables can be applied to the source code to enable simulations of higher valencies and supramolecular topologies. Moreover, *MVsim* treats the calculation of effective concentration as an additional, modular mathematical step, enabling customization with any polymer end-to-end density function. Finally, the source code is further compatible with the MATLAB curve fitting toolbox to enable parameter estimation for incompletely characterized multivalent systems.

As presented here, *MVsim* simulates interactions between systems of receptors and ligands with valencies of up to three. The choice of trivalent interactions was chosen to balance the number of computational steps needed to map the complete configurational network, which scales factorially with valency, with the ability to model complex and important instances of multivalency, such as those that occur in bispecific antibodies and the trimeric architecture of the SARS-CoV-2 S protein. Even so, the configuration nomenclature, rule-based modeling, and combinatorial computation of effective concentrations that underlie the simulations are written in the source code to accommodate all valencies and numbers of competing ligands.

While the predictive power of *MVsim* lies in its structured treatment of multivalent binding as a dynamic ensemble of microstates, this framework can create impediments toward extending the model to describe even more complex instances of multivalency. For example, the size of the configurational network increases factorially with the valency and number of molecules in the system, easily comprising thousands of rate equations and dozens of computationally-demanding PDF convolutions. Interestingly, for most multivalent systems, only a small fraction of the total network would ever be significantly populated, and thus required to accurately describe the response dynamics. To take advantage of this fact, however, requires knowing a priori the composition of this subset of the configurational network. This could be

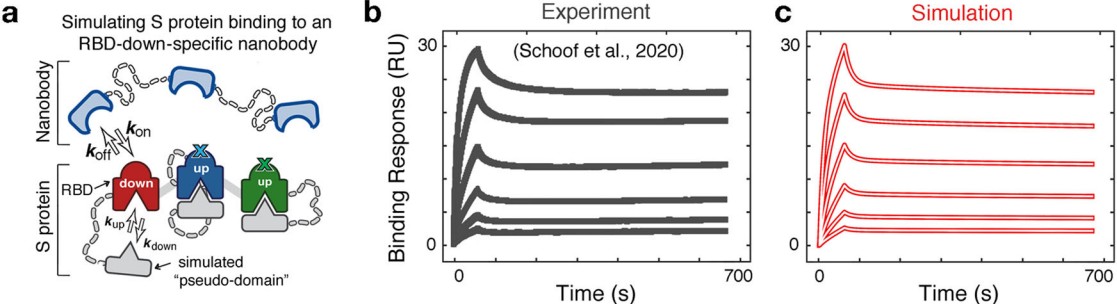

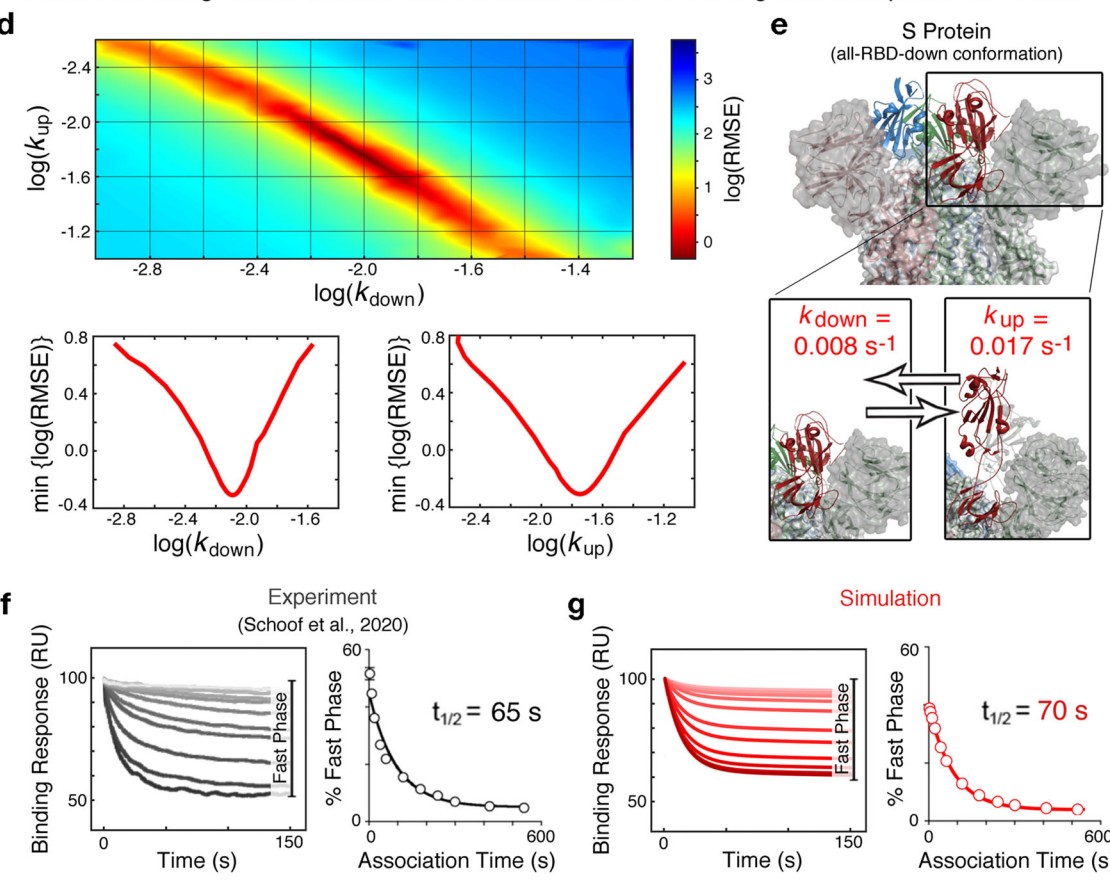

**Fig. 6 | _MVsim_ quantitatively predicts multiphasic binding response dynamics of S protein therapeutic neutralization and fits for rate constants of conformational switching. a** _MVsim_ models the conformational change as an intramolecular ligand binding event (colored in gray) that toggles the trivalent S protein between "RBD-up" and "RBD-down" conformations. The conformational change is defined by a pair of first-order kinetic rate constants $k_{down}$ and $k_{up}$. **b, c** The experimental kinetics of conformation-specific nanobody binding, adapted from ref. 37 (**b**), are qualitatively predicted by a zero-fit simulation with _MVsim_ (**c**). **d, e** Applying a gradient descent fitting routine to three of the six experimental sensorgrams in **b** ([nanobody] = 3.13, 12.5, and 50 nM; the second, fourth, and sixth curves from the bottom, respectively) that minimizes the root-mean-square error (RMSE) between model and simulation across a broad range of values for $k_{down}$ and

$k_{up}$ (**d**, top panel) converges on a unique solution for a single set of best-fit parameter values for $k_{down}$ (**d**, bottom left panel) and $k_{up}$ (**d**, bottom right panel) for an individual RBD (**e**). **f** Conformational switching half-life experiments also adapted from ref. 37, alter the relative proportion of "slow phase" inhibitor dissociation events (i.e., high-avidity bivalent and trivalent interactions) and "fast phase" inhibitor dissociation events (i.e., monovalent interactions). Here, due to relatively slow RBD switching rates, longer association times enable more S protein to be bound in high-avidity interactions, and thus give rise to small percentages of "fast phase" dissociation events. **g** To assess _MVsim_ accuracy, the fitted parameters are used in a modeling framework to simulate the experimental system and compare half-lives ($t_{1/2}$) of "RBD-up" S protein conformations. The input parameters for these simulations are given in the _MVsim_ user tutorial in the Supplementary Information.

accomplished, for example, by incorporating additional rules in the rule-based modeling routine to allow for sparse matrix sampling of the configurational microstate network and probability density functions to significantly decrease the computational time. Further, with a suitably sized set of training data, machine learning algorithms could be applied to identify input/output relationships between the structure of a given multivalent interaction system and the minimal set of rate equations

required to describe its kinetic behavior, though this data-driven approach may come at the expense of mechanistic insights. The framework of _MVsim_ can also be sensitive to cases where rate equations are biased or cannot be deterministically applied due to non-random events or low numbers of molecules, especially if these factors substantially influence the dynamics of key response-determining nodes in the configurational network.

Nonetheless, *MVsim* offers insights into important binding dynamics that are not fully represented in the current modeling landscape by quantifying all possible microstates and effective concentrations in multivalent, multispecific and/or multi-ligand systems. A simple graphical interface allows for facile simulation of user-defined systems up to trivalent-trivalent interactions and provides mechanistic insights that are not possible with more coarse-grained model descriptions. As highlighted in the examples in this study, *MVsim* has a straightforward adaptability that can address a number of important biological questions and biomedical problems involving multi-molecular interactions and offers tools to better analyze and engineer them.

## Methods

### *MVsim*

The *MVsim* multivalent simulation application was built within the MATLAB app development environment (version 2021a). Our previously reported microstate network model and odds-ratio-based calculation of effective concentration served as the foundation for creating a rule-based modeling routine for the enumeration of multivalent, multispecific, and multi-ligand-receptor binding states[23]. *MVsim* generates a model structure and all possible microstates from rules determined by the properties of the interacting molecules, including the number of binding sites on the receptors and ligands and the permissible binding configurations. All possible states and permissible transitions among these states are enumerated with a system of unique identifiers, which is detailed in the Supplementary Information.

Of the four types of parameters that *MVsim* incorporates, the association rate constant, the dissociation rate constant, and the bulk ligand concentration are directly measured or defined. The fourth type of parameter, effective ligand concentration, is computed from the three-dimensional probability density functions (PDFs) of the ligand and receptor:

$$C_{eff} = \frac{\int PDF_{ligand}(V) * PDF_{receptor}(V)dV}{constant_{normalization}}$$

where $V$ is the accessible volume and the normalization constant converts the probability into units of concentration. For a multivalent ligand or receptor, its overall PDF is represented by the convolution of PDFs of the individual binding domains and linkers that comprise it, thus incorporating biophysical properties such as domain size and linker length and rigidity. Full details of the convolution of PDFs, calculation of effective ligand concentrations, and other technical considerations in *MVsim* are provided in the Supplementary Information.

### Experimental methods

Multivalent and multispecific receptors were constructed with the C-terminal SH3 domain of the human adaptor protein Gads and the synthetic protein Prb, as used in our earlier work[23]. Multivalent and multispecific ligands incorporated the SH3-binding peptide (SBP) from the Gads cognate ligand SLP-76[42], as well as the synthetic designed Prb-binding DARPin, Pdar[53]. The valency of SLP-76 ligands was tuned without introducing significant changes in molecular weight by introducing binding-ablating alanine substitutions into individual SBP motifs. Interdomain linkers were designed to be short or long, and flexible (random coil) or rigid (alpha-helical). The DNA sequences of the receptors and ligands were synthesized as gBlocks (Integrated DNA Technologies). Receptors were introduced into pET28a (Novagen) and ligands into pMal-c5x (New England Biolabs) using standard DNA cloning methods. All proteins were recombinantly expressed in *Escherichia coli* BL21 cells, with AviTagged receptors biotinylated in vivo by co-transformation with GST-BirA. Detailed protein design methods, including the full amino acid sequences of each experimentally tested construct, are provided in the Supplementary Information and in Supplementary Table 1.

Association and dissociation kinetics between ligand and receptor constructs were quantified by surface plasmon resonance (SPR) measurements on a Biacore S200 instrument. CM5 sensor chips were first conjugated with NeutrAvidin and were then used to immobilize ~30 RUs of biotinylated receptor. In addition to this low receptor surface density, a high ligand flow rate (75 μl/min) was used to minimize mass-transfer effects. All binding measurements were performed at 25 °C in a running buffer of HBS-EP+ (10 mM HEPES, 150 mM NaCl, 3 mM EDTA, 0.05% Tween-20, pH 7.4). Additional details of the SPR experiments are provided in the Supplementary Information.

### Documentation

A full set of version release notes, instructions, user tutorial, and annotated model source code are available on the *MVsim* homepage at https://sarkarlab.github.io/MVsim/.

### Reporting summary

Further information on research design is available in the Nature Research Reporting Summary linked to this article.

## Data availability

Source data are provided with this paper.

## Code availability

The code for *MVsim* is available at https://sarkarlab.github.io/MVsim/.

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

## Acknowledgements

We thank Dr. Péter Antal for the critical review of the calculations of effective concentration in *MVsim*. This work was supported by funding from the National Laboratory of Artificial Intelligence through the National Research Development and Innovation Office under the auspices of the Ministry for Innovation and Technology and from the National Research, Development, and Innovation Fund of Hungary (TKP2021-EGA-02) (B.B.); from the National Institutes of Health (R35GM136309, R01GM113985, and R21EB022258 to C.A.S.); and from the Institute for Engineering in Medicine at the University of Minnesota (COVID-19 Rapid Response Grant to C.A.S.). The Biacore S200 instrument was made available through a shared instrument grant (S10OD021539) from the Office of Research Infrastructure Programs at the National Institutes of Health.

## Author contributions

B.B., W.J.E., and C.A.S. designed the research; B.B developed the computational approach and wrote the code for *MVsim*; W.J.E. and B.B. created the graphical user interface for *MVsim*; W.J.E. performed the surface plasmon resonance experiments; B.B. and W.J.E. conducted the *MVsim* simulations and data analyses; W.J.E., B.B., and C.A.S. wrote the paper.

## Competing interests

The authors declare no competing interests.
