## [Peer Review File · Nature Communications]

Reviewers' Comments:

Reviewer #1:

Remarks to the Author:

This is an excellent paper by Bruncsics et al. I recommend that the manuscript be reconsidered for publication after revisions. I think that a suitably revised manuscript is likely to be highly cited and very useful to researchers in the field, in diverse areas (as illustrated by the applications in the paper – synthetic biology to therapeutic design and fundamental biophysical characterization).

Specific comments:

1. In general, I found the discussion of the results in the manuscript to be insufficient. A lot of details are provided in the figure caption. Not enough information is provided to properly understand the data presented (what is being done/how was it done/ do the simulation results agree well with experiments/ how do the authors conclude/characterize whether the agreement is good, etc.). I will provide specific examples below.
2. In figure 3, how was the parametrization done? Were kinetic rate constants determined from SPR measurements? How were the pdfs for the linkers and/or the values of l_c and l_p generated?
3. As was noted in the literature more than 20 years ago, SPR-based determination of rate constants/equilibrium constants (particularly in cases where the K_d is low/binding is high-avidity) can be influenced by mass transfer limitations. Are the authors sure that their parametrization was not affected by these considerations? I refer the authors to papers by Myszka from the 1990s.
4. The data presented in Fig. 3 (e.g., Figs 3e vs. 3f; 3h vs. 3i) suggest that the curves “look” similar.
Can the comparisons be made more quantitative? It would also be useful to characterize how sensitive the “goodness of fit” is to variations in parameters. For instance, what is the effect of changing the values of l_c , l_p , k_{on} , and k_{off} by +/- 30%?
5. Fig. 4: the results need to be discussed in much greater detail in the text. For Fig. 4b, some of the trends seem off (e.g., comparison of simulation vs. data for the second bar from the right).
6. Fig. 4c: I find it difficult to understand what is being shown here.
7. The analysis of the rate of switching of RBD conformations was very nice. It would be helpful to show (maybe in the SI) how changing k_{down} and k_{up} (both by 10 fold or 0.1-fold, i.e., keeping the ratio the same) affects the simulations and their agreement with experimental data.
8. The last paragraph before the Discussion section (a discussion of applications to analyzing variants of concern) seems speculative in its current form.

Reviewer #2:

Remarks to the Author:

The authors presented a graphical user interface (GUI) for designing, predicting and even quantifying multivalent binding processes, including calculations of effective association (k_{on}) and dissociation rates (k_{off}) and binding response curves. Called as MVsim, the tool seems quite flexible and can deal with multicomponent systems, like multidomain protein or multispecific interactions between ligand and target. MVsim is based on configurational microstate network model and can be consider a follow up paper from the original work published in 2019 by the same authors at PNAS (DOI: 10.1073/pnas.1902909116). As described in the main paper, MVsim seems to be a substantial expansion from their original computational method. More importantly, the works is a nice attempt to help no-expert scientists in studies involving multivalent binding processes. After describing the default inputs and outputs of the MVsim GUI, the paper first evaluates the accuracy of the approach, showing, in my view, a reasonable agreement with binding response curves involving monovalent but also multispecific and multi-ligand interactions. The second part of the results illustrates with four examples how MVsim can also be used as molecular design and quantification tool. The final part of the paper focused on how the tool can be used to bring insights in the kinetics rates related to conformational changes SARSCoV-2 spike protein and the binding to ACE2. This section also suggests a concept that could be explored to generate new inhibitors, blocking the spike-ACE2 interactions.

Overall, the paper is well-written and seems to be a very relevant and important material for publication in the Nature Communications. However, considering that the main point of the paper is to present a GUI, which is usually made considering non-expert users, I think the authors need to address some points before the paper be fully accepted by the editor:

Major:

1) Although the description of parameter input and simulation outputs are clear, I missed some hands-one material clearly showing how to use the GUI. I did not find anything like that in all the material presented (main manuscript, SI or webpage). Sometimes, modeling/computational methods papers published in Nature Communications also indicate tutorials with a more practical view for the users (for instance, DOI: 10.1038/s41467-020-17437-5). I think one-two tutorials, or even recorded demonstrations of some examples of the paper, would be beneficial for the users. It would also strength the arguments of the authors that repeatedly say in the paper that MVsim is simple and that *“offers an intuitive and easy-to-implement molecular design toolset...”*. More importantly, the tutorial/recorded demonstration also could help for reproducibility of the data presented in the paper by other scientists.

2) In comparison to the rest of the paper, that was quite detailed, I considered the discussion section quite short. If the point is to present the GUI and a new/expanded computational method, the authors also need to present and discuss possible limitations of the approach behind the interface: configurational network model. This is extremely useful for community properly use the GUI interface.

3) In the same line as topic 2, some comparison with previous codes and approaches (like the ones pointed in references 14-22) in the discussion section would be desirable, so we could properly access how good is MVsim in relation to what has already been published before. For instance, comparison with some examples presented in previous works could be added to the SI material and mentioned in the discussion.

Minor

4) Considering the multidisciplinary audience of Nature Communications, would be desirable that some section of the paper (like introduction or even SI) presents and explain some terms in the paper, like differences between multivalent, multispecific, multi-ligand, etc. This may be minor but in sections of the paper, like the one describing Figure 3, it could be quite handy for the reader. For instance, can the multi-ligand system (Fig. 3g) also be consider a multi-specific (as shown in Fig. 3d)?

5) Here are some important small problems that can help the understanding of the figures:

- in Fig2d. "x" should be in ligand.

- in Fig4 C, left panel ... it is quite hard to read what is written in the gray boxes. Maybe an abbreviation (explained in the figure legend) could be used here with a bigger font size. The authors should double check how readable is the text in these figures. I really needed to zoom a lot to see all the details were there. For Fig 4C, even with zoom was impossible.

Reviewer #1 (Expertise: multivalent biotherapeutic design)

This is an excellent paper by Bruncsics et al. I recommend that the manuscript be reconsidered for publication after revisions. I think that a suitably revised manuscript is likely to be highly cited and very useful to researchers in the field, in diverse areas (as illustrated by the applications in the paper – synthetic biology to therapeutic design and fundamental biophysical characterization).

We thank the reviewer for this positive overall assessment.

Specific comments:

1. In general, I found the discussion of the results in the manuscript to be insufficient. A lot of details are provided in the figure caption. Not enough information is provided to properly understand the data presented (what is being done/how was it done/ do the simulation results agree well with experiments/ how do the authors conclude/characterize whether the agreement is good, etc.). I will provide specific examples below.

We agree that more explicit descriptions of the experimental data and simulation procedures will enhance the clarity of our work. Our revisions are detailed point-by-point in response to the subsequent comments.

2. In figure 3, how was the parametrization done? Were kinetic rate constants determined from SPR measurements? How were the pdfs for the linkers and/or the values of l_c and l_p generated?

Not only for Figure 3 but also for Figures 4-6, we have now revised the manuscript to clearly indicate the values and means of derivation for both the kinetic parameters (i.e., the monovalent rate constants of association, k_{on} , and dissociation, k_{off}) and the structural parameters (i.e., molecular weight, domain diameters, linker contour length, and linker persistence length).

Figure 3 simulations

2a. *We have clarified in the text (lines 190-195, 213-216, and 232-234) that the kinetic parameters used for the simulations (Fig. 3c,e,h) were determined from our own SPR measurements of the monovalent protein-protein interacting pairs (Fig. 3a).*

2b. *We have added a new figure to the Supplementary Information (Fig. S5) showing all of the parameter values as they appear as user inputs in the MVsim interface.*

2c. *To new Table S1, we have included the amino acid sequences for all of the proteins depicted in Fig. 3. Further, we have cited the literature from which we obtained the structural parameters in the main text (lines 195 and 215).*

Figure 4, 5, and 6 simulations

2d. Explanatory figure panels related to the simulations in Fig. 4, 5, and 6 have been added to the Supplementary Information (Fig. S6; Fig. S8a,b; and the accompanying MVsim User Tutorial), showing the parameter values used for these simulations as they would appear as user inputs in the MVsim interface.

2e. Additionally, the sources of these parameters are now clearly referenced in the corresponding figure legends in both the main text and Supplementary Information as being obtained either directly from the primary papers (i.e., the values for k_{on} , k_{off} , K_D , and linker contour length (l_p)) or sourced from elsewhere in the literature (i.e., the values for domain diameters, and the persistence lengths (l_p) for flexible and rigid polypeptide linkers).

3. As was noted in the literature more than 20 years ago, SPR-based determination of rate constants/equilibrium constants (particularly in cases where the K_D is low/binding is high-avidity) can be influenced by mass transfer limitations. Are the authors sure that their parametrization was not affected by these considerations? I refer the authors to papers by Myszka from the 1990s.

We thank the reviewer for raising this important point. We are indeed well aware of the seminal papers by Myszka which highlight the significant effects that mass-transfer limitations can have on SPR-based measurements, especially in experiments involving ligands with high molecular weight/low diffusivity, receptors immobilized at high densities, interactions with large association rate constants, and/or low ligand flow rates.

We have clarified and expanded upon our treatment of mass-transfer limitations with the following revisions.

3a. As now detailed in the Supplementary Information (Extended Experimental Methods), we chose interacting protein pairs with modest rate constants of association (k_{on}) and molecular weights, and we intentionally performed our SPR experiments with low immobilized surface receptor density (less than 30 RU). We also state our diagnostic use of variable flow rate to ensure our experimental system was not significantly mass-transfer limited and, for the purposes of data collection, the conservative use of high flow rates (75 μ l/min). Based upon this, we added a statement in the Fig. 3 legend to note that our kinetic parameterization was appropriately based on fits to a “rapid mixing” 1:1 Langmuir model.

3b. To ensure that potential mass-transfer limitations would not arise with changes in valency, we used a protein design approach that leveraged mutation rather than domain addition/deletion to change valency. The preparation and sequences of these proteins have been included in the Supplementary Information (Extended Experimental Methods and Table S1).

3c. As the reviewer points out, multivalency can compound the kinetics of association and dissociation through avidity. Such avidity enhancements are predominantly driven through a slowing of the effective off rate, with minimal effect due to k_{on} at the valencies studied here (Errington et al., PNAS, 2019). Multivalent avidity is thus most significantly driven through a form of statistical rebinding that reduces the apparent k_{off} . This unique feature of multivalent binding is not an artifact of the experimental setup (like mass-transfer limitations), but is instead a biologically relevant phenomenon that is explicitly treated by our network model. This is supported by our model-experiment agreement, which we have now quantified (e.g., Fig. 3c; RMSE 2.1 RU), without invoking a treatment of mass transfer.

3d. We have similar confidence that the kinetic parameters obtained from the literature (Schoof et al.; ref. 37) with which we parameterized the S protein RBD simulations (Fig. 6c) are unlikely to have been significantly affected by mass-transfer limitations due to the use of low molecular weight, monovalent nanobodies (~12 kDa), low apparent immobilized densities of S protein RBD (R_{max} ~30 RU; Supp. Info. from Schoof et al.), a measured k_{on} of modest value ($2.7 \times 10^5 \text{ M}^{-1} \text{ s}^{-1}$), and monovalent kinetics that were well-described with a simple 1:1 binding model (Fig. 3a from Schoof et al.).

4. The data presented in Fig. 3 (e.g., Figs 3e vs. 3f; 3h vs. 3i) suggest that the curves “look” similar. (A) Can the comparisons be made more quantitative? (B) It would also be useful to characterize how sensitive the “goodness of fit” is to variations in parameters. For instance, what is the effect of changing the values of I_c , I_p , k_{on} , and k_{off} by +/- 30%?

We thank the reviewer for these suggestions. Further, as the reviewer states, incorporating a quantitative measure of goodness of fit would facilitate identifying molecular parameters whose experimental values may differ from their idealized, theoretical values (e.g., I_c , I_p , k_{on} , and k_{off}).

4a. We have introduced a root-mean-square error (RMSE) assessment to quantify the goodness-of-fit between the simulations and experiments in Fig. 3. The method is described in the Supplementary Information (Extended Computational Methods, Section 2.10).

4b. We have demonstrated the utility of the RMSE algorithm by quantifying the agreement between our idealized zero-fit simulations and the experimental data (RMSE values are indicated in the Fig. 3 legend and main text descriptions on lines 199, 222, and 246).

4c. Further, as the reviewer suggests, we additionally show how zero-fit simulations in MVsim do appropriately capture the unique features of multivalent binding responses by varying I_c , I_p , and k_{on} parameter values and showing that they change the characteristic shapes of the curves and consequently the goodness of fits (Fig. S3).

5. Fig. 4: the results need to be discussed in much greater detail in the text. For Fig. 4b,

some of the trends seem off (e.g., comparison of simulation vs. data for the second bar from the right).

We have provided more details on our use of MVsim to model the experimental logic-gate system reported by Dueber et al. (2007) and our presentations in Fig. 4b and Fig. S6. To better highlight our application of MVsim to the design of these types of systems in synthetic biology, we made the following clarifying additions to the text.

5a. *First, we clarify that the published experimental study by Dueber et al. does not report an SPR-based characterization for their system, but rather uses activation of an enzymatic process to assess output. Nonetheless, MVsim can still be used to provide insights into a multivalent system evaluated through non-SPR-based experimentation (line 291-294).*

5b. *Second, as astutely noted by the reviewer, there is a lack of quantitative agreement in the trends between experiment and simulation for single-ligand inputs (particularly the yellow ligand, second bar from right), which we now better emphasize. Relatedly, we draw attention to the additional lack of agreement between experiment and simulation for the no-input condition (i.e., the leftmost bar with the grey dash symbol displays ~10% output in the experiment). We state that these points of disagreement highlight the more significant susceptibility of the experimental system, beyond what is predicted by the zero-fit simulation, toward activation in the presence of zero or one input, even though the experimental system was intended to be off in all of these cases (lines 298-301). Notably, however, we emphasize (line 296) that the simulation does predict the other undesirable one-input activations (Fig. 4b, bars ii and iii). Furthermore, the simulations extend predictions to the two-input cases, not reported in the experimental study, and reveal a worsening of the subthreshold activation that could undermine the entire design (Fig. S6b, bars v-vii; lines 294-298). We have also better labeled the plots in Fig. 4b and Fig. S6b,c to show the subthreshold conditions and the single three-input suprathreshold condition.*

5c. *Third, we propose a likely cause for the two major points of disagreement above. Given the probabilistic treatment of the simplified molecular topology in MVsim and the highly twisted configuration in the experimental system, it is likely that more significant constraints manifest in reality than are predicted by the simulations. As a result, in the absence of any input, the experimental system exists in a state resembling a mixture of ~90% “closed” conformation and ~10% “open” conformation, a situation that worsens in response to sub-threshold inputs. We better emphasize the significant steric constraints of the experimental system (Fig. 4b schematic) and the limitations of MVsim resulting from its idealized treatment of molecular structure and dynamics (line 518-522).*

5d. *Finally, we have better highlighted what we feel is the most significant capability of MVsim in this specific application. MVsim demonstrates an ability to highlight a suboptimal design element in the system (i.e., a sterically constrained topology); extrapolate this behavior to assess other potentially problematic features (e.g., significant two-input activation, which was not reported in the original experimental paper); and, finally identify areas of protein design optimization to improve system performance. Here,*

we emphasize that MVsim expands on the original report by both predicting potential incidences of subthreshold activation with single- and dual-ligand inputs, and indicating rational design optimization strategies that may enhance system performance (lines 301-309 and Fig. S6b,c).

6. Fig. 4c: I find it difficult to understand what is being shown here.

We thank the reviewer and agree that in its original form, panel C in Fig. 4 was too small, dense, and difficult to interpret.

6a. *To more clearly describe this application of MVsim without detracting from the primary emphasis in Fig. 4 of selective gating/binding, we have reorganized the main text (lines 336-344) and moved the original Fig. 4c to its own supplementary figure (Fig. S7) where it is presented in an expanded and more descriptive format.*

Here, new Fig. S7 illustrates the recognized property of multivalency to encode how multiple ligands – simultaneously encountering a common receptor – can organize themselves into a sequenced progression of dominant binding events (e.g., Ligand A then Ligand B then Ligand C despite all ligands being added to the system at once). This ordering occurs through patterns of cooperative, competitive interaction that are not accessible to monovalent counterparts. We demonstrate how MVsim can be used for the exploration and application of this kinetic programmability.

7. The analysis of the rate of switching of RBD conformations was very nice. It would be helpful to show (maybe in the SI) how changing k_{down} and k_{up} (both by 10 fold or 0.1-fold, i.e., keeping the ratio the same) affects the simulations and their agreement with experimental data.

We thank the reviewer for the critical insight as to whether the parameter values of k_{down} and k_{up} are uniquely determined by the multiphasic curve shapes.

7a. *To address this important point, we used our new RMSE calculator in a Bayesian optimization algorithm (Supplementary Information, Extended Computational Methods) to show that fitting the system of differential equations for k_{down} and k_{up} indeed converges on an optimal set of values for these first-order rate constants ($k_{down} = 0.008 \text{ s}^{-1}$ and $k_{up} = 0.017 \text{ s}^{-1}$; Fig. 6d). These quantitatively fitted parameter values differ slightly from the previous values derived from qualitative visual inspection of the goodness of fits ($k_{down} = 0.006 \text{ s}^{-1}$ and $k_{up} = 0.011 \text{ s}^{-1}$), but notably, the k_{down}/k_{up} ratio remains largely unchanged. This ratio constraint is visually evident in the heat map in Fig. 6d (with a color palette on a log scale), so even fitting by eye would be expected to lead to approximately correct values. Nevertheless, we sincerely appreciate the reviewer's earlier suggestion to quantify the goodness of fit, and we are now able to report quantitatively fitted values.*

7b. *For better visualization of the agreement between the experimental and simulated*

kinetic traces, we show how changing k_{down} and k_{up} by 4-fold in either direction, while maintaining the ratio, leads to evidently poor fits (Fig. S9). (Based on Fig. 6d, changing these parameters by a full order of magnitude was unnecessary to demonstrate this.)

8. The last paragraph before the Discussion section (a discussion of applications to analyzing variants of concern) seems speculative in its current form.

We agree with the reviewer that this speculative paragraph is more appropriate for the Discussion section where we discuss areas of future direction and development.

8a. *We have re-written this paragraph and included it in the Discussion (lines 476-481).*

Reviewer #2 (Expertise: antibody design, ML)

The authors presented a graphical user interface (GUI) for designing, predicting and even quantifying multivalent binding processes, including calculations of effective association (k_{on}) and dissociation rates (k_{off}) and binding response curves. Called as MVsim, the tool seems quite flexible and can deal with multicomponent systems, like multidomain protein or multispecific interactions between ligand and target. MVsim is based on configurational microstate network model and can be considered a follow up paper from the original work published in 2019 by the same authors at PNAS (DOI: 10.1073/pnas.1902909116). As described in the main paper, MVsim seems to be a substantial expansion from their original computational method. More importantly, the work is a nice attempt to help non-expert scientists in studies involving multivalent binding processes. After describing the default inputs and outputs of the MVsim GUI, the paper first evaluates the accuracy of the approach, showing, in my view, a reasonable agreement with binding response curves involving monovalent but also multispecific and multi-ligand interactions. The second part of the results illustrates with four examples how MVsim can also be used as molecular design and quantification tool. The final part of the paper focused on how the tool can be used to bring insights in the kinetics rates related to conformational changes SARS-CoV-2 spike protein and the binding to ACE2. This section also suggests a concept that could be explored to generate new inhibitors, blocking the spike-ACE2 interactions.

Overall, the paper is well-written and seems to be a very relevant and important material for publication in the Nature Communications. However, considering that the main point of the paper is to present a GUI, which is usually made considering non-expert users, I think the authors need to address some points before the paper be fully accepted by the editor:

We appreciate this positive overview. And while we agree that the MVsim GUI is an essential component of this work to maximize accessibility for non-expert users, we thank the reviewer for also recognizing that the underlying computational method represents a significant advance from our previous work.

Major:

1. Although the description of parameter input and simulation outputs are clear, I missed some hands-on material clearly showing how to use the GUI. I did not find anything like that in all the material presented (main manuscript, SI or webpage). Sometimes, modeling/computational methods papers published in Nature Communications also indicate tutorials with a more practical view for the users (for instance, DOI: 10.1038/s41467-020-17437-5). I think one-two tutorials, or even recorded demonstrations of some examples of the paper, would be beneficial for the users. It would also strengthen the arguments of the authors that repeatedly say in the paper that MVsim is simple and that “*offers an intuitive and easy-to-implement molecular design toolset...*”. More importantly, the tutorial/recorded demonstration also could help for reproducibility of the data presented in the paper by other scientists.

We thank the reviewer for highlighting the need to maximize usability of MVsim by providing users with instructional materials to guide their practical interactions with the GUI.

1a. *We have updated the GUI with interactive “mouseover” labeling that clearly indicates what each of the parameter fields is (e.g., contour length, persistence length, and domain diameter).*

1b. *For each simulation that we present in Figures 3 through 6 we have now added new information to the Supplementary Information (Fig. S5 and MVsim User Tutorial), showing the parameter values that were used in the simulations as they would appear in the GUI. These supplementary figures should enable all users to easily reproduce and/or adapt our simulations. Callouts to each of these supplementary figures or tutorials were added to the corresponding figure legends.*

1c. *We have created walk-through tutorials that are included in both the Supplementary Information (MVsim User Tutorial) and on the MVsim GitHub webpage. The first tutorial guides users through the molecular design/parameterization input and output interfaces, as extensions of what is displayed in Figs. 1 and 2. The second tutorial takes the user through the series of steps to perform the more elaborate SARS-CoV-2 S protein simulations that are presented in Figures 5 and 6.*

2. In comparison to the rest of the paper, that was quite detailed, I considered the discussion section quite short. If the point is to present the GUI and a new/expanded computational method, the authors also need to present and discuss possible limitations of the approach behind the interface: configurational network model. This is extremely useful for community properly use the GUI interface.

We agree that these points about the computational method should be more clearly articulated in the Discussion.

2a. *We have expanded the Discussion to include an assessment of the ways in which the structure, scope, and implementations of the MVsim network model are constrained and/or idealized; further, we discuss the types of molecular systems that may be poorly predicted by MVsim with zero-fit/ab initio simulations (lines 502-522).*

3. In the same line as topic 2, some comparison with previous codes and approaches (like the ones pointed in references 14-22) in the discussion section would be desirable, so we could properly assess how good is MVsim in relation to what has already been published before. For instance, comparison with some examples presented in previous works could be added to the SI material and mentioned in the discussion.

We appreciate the importance of explicitly placing MVsim within the spectrum of existing approaches for modeling multivalent systems.

3a. We have further expanded the Introduction and Discussion to include a survey of existing modeling approaches, and how MVsim is positioned among them (lines 39-60 and 447-457). Finally, we include a concluding discussion describing future extensions that are compatible with our framework and which would enable MVsim to be current and stay apace with the continual advances in multivalent design (lines 510-518).

Minor

4. Considering the multidisciplinary audience of Nature Communications, would be desirable that some section of the paper (like introduction or even SI) presents and explain some terms in the paper, like differences between multivalent, multispecific, multi-ligand, etc. This may be minor but in sections of the paper, like the one describing Figure 3, it could be quite handy for the reader. For instance, can the multi-ligand system (Fig. 3g) also be consider a multi-specific (as shown in Fig. 3d)?

We thank the reviewer for identifying the need to clarify our use of terminology describing molecular system design. As the reviewer correctly indicates, these terms are not mutually exclusive, which we acknowledge could be confusing to the uninitiated reader. For example, all multispecific systems are multivalent but not vice versa; additionally, the particular multi-ligand system we show in Fig. 3g is indeed also multispecific (and multivalent).

4a. We have included important definitions in the Introduction (lines 76-80). Additionally, we clarify the relevant terminology when describing the systems that appear in the Results section (e.g., lines 186-190) and in the figure captions.

5. Here are some important small problems that can help the understanding of the figures:
- in Fig2d. "x" should be in ligand.
- in Fig4 C, left panel ... it is quite hard to read what is written in the gray boxes. Maybe an abbreviation (explained in the figure legend) could be used here with a bigger font size. The authors should double check how readable is the text in these figures. I really needed to zoom a lot to see all the details were there. For Fig 4C, even with zoom was impossible.

We thank the reviewer for alerting us to the typographical errors and issues with figure format legibility.

5a. We have corrected the error in Fig. 2d by moving the "x" to the ligand field.

5b. To enhance the legibility and clarity of Fig. 4, we have moved panel 4c to the Supplementary Information (Fig. S7 where we have remade it as a larger, stand-alone figure) and commensurately increased the size of the main text Fig. 4. In the new Fig. S7 (accompanied by main text lines 336-344), we aim to more clearly communicate the application of MVsim as a guide for establishing ordered binding events in multi-ligand cascades through competitive interactions.

Reviewers' Comments:

Reviewer #1:

Remarks to the Author:

My concerns have been addressed. I believe that the manuscript is suitable for publication in its current form.

Reviewer #2:

Remarks to the Author:

The authors properly answered all my concerns. So, now, I fully support and recommend this beautiful manuscript for publication in Nature Communications.

Reviewer #3:

Remarks to the Author:

The authors have described a newly developed tool, MVsim for the analysis of multivalent interactions. Based on MATLAB, GUI was developed for non-experts to analyze and characterize the mono and multivalent binding interactions. While the multi- to multi-domain binding become complicated because of the increasing number of binding combination, MVsim automatically emulates all possible binding states to simulate the binding kinetics. Experimental data and simulation fit very well. The fits were evaluated quantitatively, indicating the accuracy of the model. MVsim was also used not only to analyze the SPR data but also to analyze and fit the catalytic activity data of the multidomain proteins. Furthermore, the effect of a domain sequence, linker length, and linker flexibility on the binding was evaluated quantitatively. While the simulation matched well with intuition, it is important that the effects could be assessed quantitatively. Finally, the interaction between ACE2 and S protein was investigated. MVsim was useful in designing inhibitor proteins and understanding the role of conformational change of S protein on the binding of nanobody.

I find the research very interesting and MVsim a very useful tool. I downloaded and used MVsim for simulating some sets of molecular systems. Download and installation of the App were straightforward and easy. I was also able to run simulations easily with various molecular systems. In sum, I think this App is easy enough for the non-experts to use for their own research. Multiple examples described show that MVsim is indeed a useful tool to analyze and understand multivalent protein binding.

Minor comments:

- Some researchers might also want to do a batch analysis where multiple inputs want to be simulated. Is it easy to use the MVsim for this purpose?
- Because the paper already has been evaluated by two reviewers, I read the point-by-point replies to the two reviewers. The authors have fully replied to the reviewer's concerns. One very minor comment is the size of the letters in the Figures. One of the reviewers raised the same concern, however, I still have difficulty seeing the words in the Figures. For example, " Fig 4b, N-WASP, SH3-binding peptides" are still difficult to see.

In summary, I suggest the paper be published in Nature Communications.

Reviewer #1 (Remarks to the Author)

My concerns have been addressed. I believe that the manuscript is suitable for publication in its current form.

We thank the reviewer for the constructive comments during the review process and for supporting publication of our manuscript.

Reviewer #2 (Remarks to the Author)

The authors properly answered all my concerns. So, now, I fully support and recommend this beautiful manuscript for publication in Nature Communications.

We thank the reviewer for the constructive comments during the review process and for supporting publication of our manuscript.

Reviewer #3 (Remarks to the Author)

The authors have described a newly developed tool, MVsim for the analysis of multivalent interactions. Based on MATLAB, GUI was developed for non-experts to analyze and characterize the mono and multivalent binding interactions. While the multi- to multi-domain binding become complicated because of the increasing number of binding combination, MVsim automatically emulates all possible binding states to simulate the binding kinetics. Experimental data and simulation fit very well. The fits were evaluated quantitatively, indicating the accuracy of the model. MVsim was also used not only to analyze the SPR data but also to analyze and fit the catalytic activity data of the multidomain proteins. Furthermore, the effect of a domain sequence, linker length, and linker flexibility on the binding was evaluated quantitatively. While the simulation matched well with intuition, it is important that the effects could be assessed quantitatively. Finally, the interaction between ACE2 and S protein was investigated. MVsim was useful in designing inhibitor proteins and understanding the role of conformational change of S protein on the binding of nanobody.

I find the research very interesting and MVsim a very useful tool. I downloaded and used MVsim for simulating some sets of molecular systems. Download and installation of the App were straightforward and easy. I was also able to run simulations easily with various molecular systems. In sum, I think this App is easy enough for the non-experts to use for their own research. Multiple examples described show that MVsim is indeed a useful tool to analyze and understand multivalent protein binding.

We appreciate this positive summary of our manuscript, and we thank the reviewer for highlighting the ease of using MVsim as well as its utility in various applications.

Minor comments:

Some researchers might also want to do a batch analysis where multiple inputs want to be simulated. Is it easy to use the MVsim for this purpose?

We recognize the utility of this feature, so we thank the reviewer for raising this point. MVsim previously did accommodate batch analysis by exporting and overlaying individual simulations, which we now explicitly highlight in our user tutorial (slides 34-41). Furthermore, we have updated MVsim to automate the most common form of batch analysis in SPR experiments – multiple input ligand concentrations (slide 47 in the user tutorial). Other types of batch analysis are possible with straightforward customization of the code.

Because the paper already has been evaluated by two reviewers, I read the point-by-point replies to the two reviewers. The authors have fully replied to the reviewer's concerns. One very minor comment is the size of the letters in the Figures. One of the reviewers raised the same concern, however, I still have difficulty seeing the words in the Figures. For example, " Fig 4b, N-WASP, SH3-binding peptides" are still difficult to see.

We have revised the figures to improve legibility.

In summary, I suggest the paper be published in Nature Communications.

We thank the reviewer for supporting publication of our manuscript.